# Report from the Western Canadian Gastrointestinal Cancer Consensus Conference Virtual Education Series—Transition from Local to System Therapy and Optimal Sequencing of Systemic Therapy for HCC

**Adnan Zaidi** [1,*], **Shahid Ahmed** [1], **Shahida Ahmed** [2], **Bryan Brunet** [1], **Janine Davies** [3], **Corinne Doll** [4], **Dorie-Anna Dueck** [1], **Vallerie Gordon** [2], **Pamela Hebbard** [2], **Christina Kim** [2], **Duc Le** [1], **Richard Lee-Ying** [4], **Howard Lim** [3], **Dave Liu** [5], **John Paul McGhie** [6], **Karen Mulder** [7], **Jason Park** [2], **Daniel Renouf** [3], **Devin Schellenberg** [8], **Ralph P. W. Wong** [2] and **Mike Moser** [9]

1 Saskatoon Cancer Center, Saskatchewan Cancer Agency, Saskatoon, SK S7N 4H4, Canada; shahid.ahmed@saskcancer.ca (S.A.); Bryan.Brunet@saskcancer.ca (B.B.); Dorie-Anna.Dueck@saskcancer.ca (D.-A.D.); Duc.le@saskcancer.ca (D.L.)

2 CancerCare Manitoba, Winnipeg, MB R3E 0V9, Canada; sahmed1@cancercare.mb.ca (S.A.); vgordon1@cancercare.mb.ca (V.G.); phebbard@cancercare.mb.ca (P.H.); Ckim3@cancercare.mb.ca (C.K.); JPARK@manitoba-physicians.ca (J.P.); rwong2@cancercare.mb.ca (R.P.W.W.)

3 British Columbia Cancer Agency, Vancouver, BC V5Z 4E6, Canada; jan.davies@bccancer.bc.ca (J.D.); hlim@bccancer.bc.ca (H.L.); drenouf@bccancer.bc.ca (D.R.)

4 Tom Baker Cancer Center, Alberta Health Service, Calgary, AB T2N 4H2, Canada; Corinne.Doll@albertahealthservices.ca (C.D.); Richard.Lee-Ying@albertahealthservices.ca (R.L.-Y.)

5 Vancouver Coastal Health, Vancouver, BC V6T1Z3, Canada; dave.liu@vch.ca

6 British Columbia Cancer Agency, Victoria, BC V8R 6V5, Canada; jmcghie@bccancer.bc.ca

7 Cross Cancer Institute, Alberta Health Services, Edmonton, AB T6G 1Z2, Canada; Karen.Mulder@albertahealthservices.ca

8 British Columbia Cancer Agency, Surrey, BC V3V 1Z2, Canada; dschellenberg@bccancer.bc.ca

9 Department of Surgery, School of Medicine, University of Saskatoon, Saskatoon, SK S7N 0W8, Canada; mam305@mail.usask.ca

* Correspondence: adnan.zaidi@saskcancer.ca; Tel.: +1-(306)-655-2710

**Abstract:** The Western Canadian Gastrointestinal Cancer Consensus Conference (WC-5) convened virtually on 10 February 2021. The WC-5 is an interactive multidisciplinary conference attended by health care professionals from across Western Canada (British Columbia, Alberta, Saskatchewan, and Manitoba) who are involved in the care of patients with gastrointestinal cancer. Surgical, medical, and radiation oncologists; pathologists; radiologists; and allied health care professionals participated in presentation and discussion sessions for the purpose of developing the recommendations presented here. This consensus statement addresses current issues in the management of hepatocellular cancer (HCC). Recommendations have been made for the transition from local to systemic therapy and the optimal sequencing of systemic regimens in the management of HCC.

**Keywords:** hepatocellular cancer; local therapy; TACE; SABR; systemic therapy; immunotherapy

## 1. Terms of Reference

### 1.1. Purpose

The aim of the Western Canadian Gastrointestinal Cancer Consensus Conference is to develop consensus opinions of oncologists and allied health professionals from across Western Canada with respect to best care practices and improving care and outcomes for patients with gastrointestinal cancers.

## 1.2. Participants

The Western Canadian Gastrointestinal Cancer Consensus Conference welcomes medical oncologists, radiation oncologists, surgical oncologists, pathologists, radiologists, gastroenterologists, and allied health professionals from Western Canada who are involved in the care of patients with gastrointestinal malignancies (Table 1).

**Table 1.** Attendees.

| | | | |
|---|---|---|---|
| Adnan Zaidi | Medical Oncologist | Saskatchewan Cancer Agency | SK |
| Adrian Bak | Gastroenterologist | Kelowna Gastroenterology | BC |
| Andrew McKay | Surgeon | U of MB | MB |
| Asif Shaikh | Medical Oncologist | BC Cancer | BC |
| Brady Anderson | Fellow | CancerCare Manitoba | MB |
| Christina Kim | Medical Oncologist | CancerCare Manitoba | MB |
| Corinne Doll | Radiation Oncologist | University of Calgary | AB |
| Debiprasad Papu Tripathy | Hepatologist | University of Saskatchewan | SK |
| Delia Sauciuc | Medical Oncologist | BC Cancer | BC |
| Dorie-Anna Dueck | Medical Oncologist | Saskatchewan Cancer Agency | SK |
| Duc Le | Radiation Oncologist | Saskatoon Cancer Centre | SK |
| Edward Hardy | Medical Oncologist | BC Cancer—Interior Health Authority | BC |
| Elvira Planincic | Clinic Nurse | Cancer Care Manitoba | MB |
| Gavin Beck | HPB Surgeon | University of Saskatchewan | SK |
| Hatim Karachiwala | Medical Oncologist | Cross Cancer Institute | AB |
| Hongwei Liu | Radiation Oncologist | Central Alberta Cancer Center | AB |
| Howard Lim | Medical Oncologist | BC Cancer | BC |
| Jacob Easaw | Medical Oncologist | Cross Cancer Institute | AB |
| Janine Davies | Medical Oncologist | BC Cancer | BC |
| Jennifer Spratlin | Medical Oncologist | Cross Cancer Institute | AB |
| Jiti Gill | Medical Oncologist | BC Cancer | BC |
| JP McGhie | Medical Oncologist | BC Cancer | BC |
| Junliang Liu | Radiation Oncologist | CancerCare Manitoba | MB |
| Karen King | Medical Oncologist | Cross Cancer Institute | AB |
| Karen Mulder | Medical Oncologist | Cross Cancer Institute | AB |
| Keith Tankel | Radiation Oncologist | Cross Cancer Institute | AB |
| Kelly Cheung | Pharmacist | CancerCare Manitoba | MB |
| Kim Paulson | Radiation Oncologist | Cross Cancer Institute | AB |
| Kimberly Hagel | Medical Oncologist | Sask Cancer Agency | SK |
| Kurian Joseph | Radiation Oncologist | Cross Cancer Institute | AB |
| Lyly Le | Medical Oncologist | BC Cancer | BC |
| Maged Nashed | Radiation Oncologist | CancerCare Manitoba | MB |
| Marianne Krahn | Medical oncologist | CancerCare Manitoba | MB |
| Marie Moreau | Oncologist? | Cancer Care? | AB |
| Mark Kristjanson | Community Oncology Program | CancerCare Manitoba | MB |
| Mike Moser | HBP Surgeon | University of Saskatchewan | SK |
| Muhammad Zulfiqar | Medical Oncologist | BC Cancer Agency | BC |
| Mussawar Iqbal | Medical Oncologist | Allan Blair Cancer Centre | SK |
| Omar Abdelsalam | Physician? | Cross Cancer Institute | AB |
| Osama Ahmed | Medical Oncologist | Saskatoon Cancer Centre | SK |
| Rebekah Rittberg | Resident | CancerCare Manitoba | MB |
| Sangjune Lee | Radiation Oncologist | Tom Baker Cancer Centre | AB |
| Shahid Ahmed | Medical Oncologist | Saskatchewan Cancer Agency | SK |
| Shahida Ahmed | Radiation Oncologist | CancerCare Manitoba | MB |
| Sharlene Gill | Medical Oncologist | BC Cancer Agency | BC |
| Sheryl Koski | Medical Oncologist | Cross Cancer Institute | AB |
| Shilo Lefresne | Radiation Oncologist | BC Cancer | BC |
| Stephanie Lelond | Clinical Nurse Specialist | CancerCare Manitoba | MB |
| Stephen Congly | Transplant Hepatologist | University of Calgary | AB |
| Tirath Nijjar | Internal Medicine | Cross Cancer Institute | AB |
| Vallerie Gordon | Medical Oncologist | CancerCare Manitoba | MB |
| Vincent Tam | Medical Oncologist | Tom Baker Cancer Centre | AB |
| Wei Xiong | Associate Professor | UBC | BC |
| Zoe Ignacio | Research Nurse | CancerCare Manitoba | MB |
| Dave Liu | Interventional Radiologist | Vancouver General Hospital | BC |
| Devin Schellenberg | Radiation Oncologist | BC Cancer—Surrey | BC |
| Ralph Wong | Medical Oncologist | CancerCare Manitoba | MB |

*1.3. Target Audience*

The recommendations presented here are targeted towards health care professionals involved in the care of patients with hepatocellular cancer (HCC).

*1.4. Basis of Recommendations*

The recommendations published here are based on presentations and discussions of the best available evidence. Where applicable, references are cited.

## 2. Question 1

*2.1. When Should We Transition from Local to Systemic Therapy for HCC?*

Recommendations

- Patients should be reviewed in a multidisciplinary fashion to determine the eligibility and sequencing of therapies;
- Liver status should be a Child–Pugh class A score to be considered for systemic therapy;
- Systemic therapy should be considered in patients with extrahepatic disease;
- In patients with localized disease, systemic therapy should be considered in patients where local regional therapy has failed, or patient is not eligible. The assessment of disease being considered refractory or ineligible should be made by a multidisciplinary consensus;
- In addition, systemic therapy should be considered early in cases where local therapy is unlikely to be beneficial and could be associated with an increased risk of deterioration of liver function from Child–Pugh class A to class B, such as in patients with bulky disease.

This is also summarized in Figure 1, a local–regional therapy flow chart.

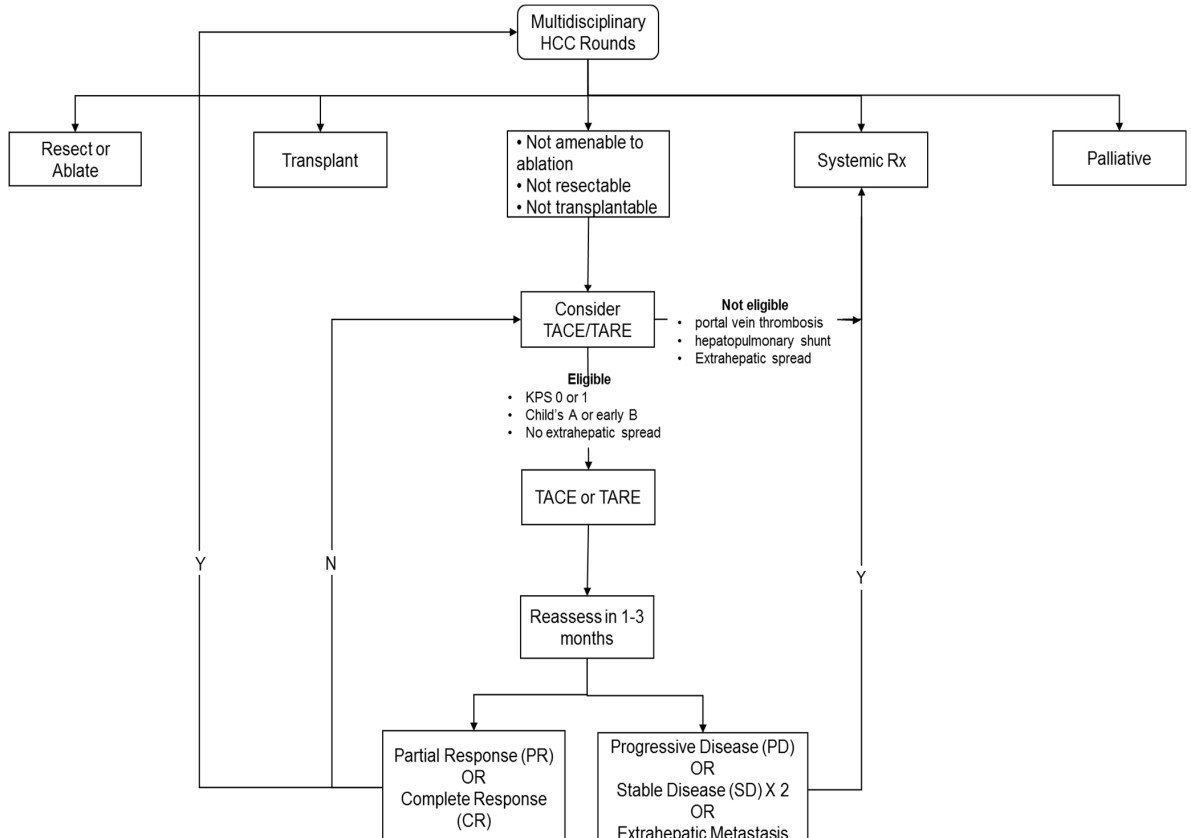

**Figure 1.** Local–regional therapy flow chart.

*2.2. Summary of Evidence*

Resection or liver transplantation, when possible, are associated with the best chance of long-term survival in patients with HCC, with five-year survival rates ranging between 40% and 70% for resection and 52% and 82% for transplantation [1]. However, resection is only feasible in patients with relatively small tumor volume, preserved liver function (Child–Pugh class A), and excellent performance status. Transplantation as a treatment option is limited by patient comorbidities, age, and the availability of donor organs. As such, only a minority of patients with HCC will be amenable to resection or transplantation, estimated as 25–37% [2].

Fortunately, there are currently more non-surgical options than ever for the locoregional treatment of HCC, including radiofrequency and microwave ablation [3], transarterial chemoembolization (TACE), and bland embolization. More recent treatment options, including focused external beam radiation–stereotactic ablative radiotherapy (SABR), also known as stereotactic body radiation therapy (SBRT) [4,5], and radioembolization (TARE and SIRT) and radiation lobectomy [6], are showing increasingly promising results.

The main difference among the modalities is that radiofrequency, microwave and SABR are treatments that aim to eradicate lesions, whereas TACE and TARE are generally more palliative and rarely result in complete tumor eradication.

### 2.2.1. Radiation-Based Treatments

With the establishment of complex treatment planning software and image guidance over the last 20 years, stereotactic ablative body radiotherapy (SABR) has become a mechanism to deliver tightly focused, tumor-eradicating doses of radiation to hepatocellular cancer while sparing the surrounding normal liver. This treatment involves a combination of the spatial precision and administration of tumoricidal, external beam doses of radiation. In multiple single- and multi-institution case series, SABR has resulted in excellent local control (>80% at 3 years [7,8] for HCC lesions. Although there have not been any completed head-to-head trials of SABR vs. radiofrequency ablation (RFA) or surgery, SABR is generally reserved for tumors that would be sub-optimally treated with RFA, such as: (1) tumors difficult to access at the dome of the diaphragm, adjacent to the heart or in the caudate lobe; (2) those near a large blood supply where a heat sink would reduce RFA effectiveness; or (3) larger tumors (>3–5 cm) where multiple ablations would be needed to cover the entire tumor volume [9].

SABR is a versatile tool in multidisciplinary treatment pathways of HCC. It can be delivered safely in an outpatient setting and is well tolerated by both elderly patients and those whose comorbidities make them poor surgical candidates. SABR can be used safely after previous RFA failure for local tumor recurrence. It can also be added to TACE if there are only 1–3 sites requiring eradication. Furthermore, it can usually be delivered safely after previous surgical resections (i.e., in a small remaining liver volume), and should disease progress post-SABR, this would not preclude further treatment with TACE, RFA or a systemic agent.

Typically, it is not used for multifocal lesions where TACE or TARE are better suited to treat lobar or widespread distributions of disease. This makes the combination of SABR with TACE or with systemic agents intellectually appealing, because local eradication could be combined with reductions in regional and/or systemic risks. Combination therapy trials will hopefully be forthcoming in the multidisciplinary management of HCC.

### 2.2.2. TACE and TARE

The majority of published reports on the non-surgical locoregional treatment of HCC are in the area of TACE with drug-eluting beads or lipiodol. The blood supply of HCCs is predominantly from the hepatic artery, embolization via the hepatic artery is particularly appealing as a treatment modality; in general, the embolized particles are taken up efficiently by the vascular HCC tumors. A large meta-analysis has confirmed that TACE is beneficial to patients with intermediate-stage HCC [10]

SIRT compares favorably to TACE, but has such advantage that it may be used in patients with portal vein thrombosis, among others [11,12]. The mechanism of action of SIRT is the targeted delivery of radioactive compounds via the hepatic artery branches [13]. This is in contrast to TACE, where there is a focused delivery of chemotherapy, and at the same time, an occlusion of small arterioles feeding the tumor. Due to the ischemia produced, super-selective TACE is recommended wherever possible, and this approach has led to significantly better overall survival than non-super-selective TACE [14]. A few studies have suggested that bland embolization may be equivalent to TACE [15]. This suggests that the effect of TACE may be due in large part to the occlusion of the arterial blood supply to the tumor.

A major disadvantage of TACE is that repeated embolic treatments lead to a decrease in liver function. Even with super-selective delivery of the beads, there is no doubt of incurring injury and necrosis to normal liver tissue surrounding the tumor. Deterioration in liver function has been documented with repeated TACE [16], with even worse deterioration seen for patients outside of the up-to-seven criteria [17]. (If the sum of the diameters of the HCC plus the number of HCCs is greater than 7, this predicts worse outcomes). Furthermore, the response rates to TACE decrease with each subsequent TACE treatment, as reported in the OPTIMIS study (the outcomes of HCC patients treated with TACE and followed by early, not early, or not at all sorafenib) [18]. This may be related to the loss of small arterioles resulting from prior treatments, leading to less efficient bead delivery on subsequent TACE treatments.

Deciding the best time to transition a patient from locoregional therapies to systemic therapies is necessarily a balancing act between the response to TACE and the maintenance of liver function. Furthermore, the use of the new systemic therapy regimens, including atezolizumab and bevacizumab, or the older first-line option sorafenib, is limited to patients with good liver function (Child–Pugh class A). As such, bevacizumab, atezolizumab, or sorafenib cannot be recommended in patients with Child–Pugh class B or C cirrhosis.

Thus, the timing of the transition from locoregional therapy to systemic therapy involves striking a balance between the beneficial effects of TACE and the decrease in liver function from repeated treatments that may make the patient ineligible to proceed with systemic treatment using newer agents.

Extensive retrospective studies have demonstrated improved outcomes from using multi-disciplinary tumor boards to discuss HCC cases [19,20]. Therefore, whenever possible, patients should be reviewed in a multi-disciplinary setting to determine appropriate treatments, and the timing and sequencing of therapies. Patients who have been started on systemic therapy and are responding to this treatment can be rediscussed at multidisciplinary tumor boards, and the role of local therapy could be revisited.

### 2.2.3. Indications for TACE

The place of TACE in the management of hepatocellular carcinoma is, in general, those patients considered to be unresectable or untransplantable, but also with good performance status, reasonable liver function (Child–Pugh classification A or early B), and no evidence of metastatic (extrahepatic) disease.

### 2.2.4. Ineligibility for TACE

As discussed above, TACE relies both upon focused chemotherapy drug delivery, as well as the occlusion of arterioles feeding the tumor. After arterial embolization treatment, the nearby liver parenchyma is sustained with portal venous blood flow. Complete portal vein thrombosis is, therefore, a contraindication to the use of TACE because this would lead to infarction and possible abscess of the treated area. The presence of a hepato-pulmonary shunt is also considered to be an absolute contraindication to the initiation or continuation of TACE treatments.

Portal vein invasion to an extent less than total thrombosis has been investigated in several retrospective studies to see if TACE is still feasible and beneficial or if these patients

should preferably be treated with systemic treatment [21,22]. A recent meta-analysis looking at the results of 11 studies with portal vein tumor invasion concluded that there was an overall survival benefit in using sorafenib compared to TACE treatments in this subset of patients [23].

### 2.2.5. TACE Progression

Follow-up imaging should be performed 1–3 months after each TACE treatment. The follow-up images should then be assessed for response to treatment and discussed in a multi-disciplinary team setting. A common way to assess treatment response that has been validated in HCC is the mRECIST (modified Response Evaluation Criteria in Solid Tumors) [24].

Interval development of extrahepatic disease or significant portal vein invasion on imaging or reports of deteriorating liver function are the key reasons to halt further TACE treatments and refer to systemic therapy instead. Other factors, such as tumor bulk and liver function, are difficult to define objectively, and instead can serve as debate points in multi-disciplinary discussions.

### 2.2.6. TACE Refractoriness

Although there has been general agreement that TACE refractoriness should be an indication for the transition to systemic therapy, a definition of TACE refractoriness was elusive for a number of years. Choi et al. studied 200 patients with HCC beyond the Milan criteria for liver transplantation [25]. About one-quarter of patients showed no objective response following two TACE sessions. Twenty-eight of these patients, or just over half of the non-responders, nonetheless received further TACE treatments. This group was noted to have only a 10.7% objective response rate and markedly poorer survival than the group with an objective response.

Another study reported similar results involving 265 patients with advanced HCC who were treated with TACE followed by sorafenib [26]. Overall survival was significantly longer in the subgroup of patients treated with two or fewer ineffective TACE procedures before switching TACE to sorafenib, and survival was significantly worse in those patients undergoing three or more consecutive ineffective TACE procedures.

Although this has not been subjected to randomized controlled trial, if a patient has two successive TACE treatments and the follow-up imaging does not show an objective response following either treatment, it seems unlikely that there will be a response with subsequent treatments. Therefore, based on the indirect evidence noted above, in patients showing no response to two successive TACE treatments, a referral for systemic treatment should be considered.

Each additional TACE procedure increases the chance of diminished liver function and decreases the chance of exhibiting an objective response; therefore, the onus should be on the multi-disciplinary team to provide strong justification for the next TACE treatment.

As a final point, a small percentage of patients with intermediate-stage HCC will exhibit a complete response following TACE. The possibility that a patient who was previously considered unresectable can become resectable or transplantable after a successful TACE treatment should not be overlooked; all patients exhibiting a treatment response should be regularly reassessed by the team for the possibility of resection or transplantation.

Many TACE treatments in the past were likely ordered beyond the point when they were beneficial to the patients. It is hoped that with the increase in multi-disciplinary teams and the advent of newer, more successful systemic therapies, the transition from locoregional therapy to systemic therapy will be considered sooner than it has been in the past. This should lead to improved results in the coming decade for our patients with HCC.

## 3. Question 2

*3.1. What Is the Optimal Sequencing of Systemic Therapy for HCC?*

- The optimal sequence of systemic therapy in patients with advanced HCC is evolving, and it is determined by patient- and disease-related factors and access to novel compounds. Enrollment in clinical trials should be considered where possible;
- Atezolizumab and bevacizumab should be considered as standard first-line therapy in appropriate patients. For this group of patients, second line treatment could be lenvatinib or sorafenib and third-line therapy with cabozantinib or regorafenib. If regorafenib is given in the third line, fourth-line therapy can be with cabozantinib;
- In patients who are not appropriate for or who decline atezolizumab and bevacizumab, first-line therapy with lenvatinib or sorafenib is appropriate. Second-line therapy for this group can be with cabozantinib or regorafenib. If regorafenib is given in the second line, third-line therapy can be with cabozantinib;
- In patients that have not received immunotherapy or have poor tolerance of a tyrosine kinase inhibitor, single-agent immunotherapy has provided a modest survival benefit and could be considered.

This is also summarized in Figure 2, a systemic therapy flow chart.

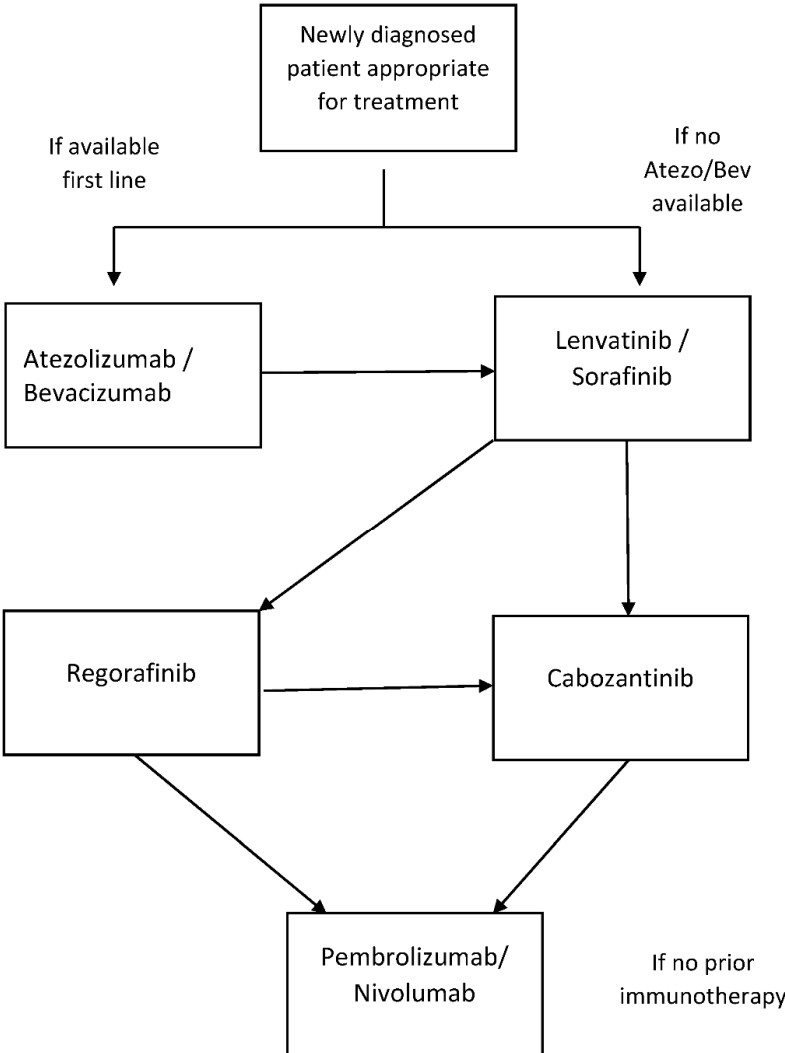

**Figure 2.** Systemic therapy flow chart.

### 3.2. Summary of Evidence

The treatment for hepatocellular carcinoma (HCC) is very often dictated by underlying liver disease. It is usually diagnosed in patients with chronic liver disease and treatment options depend on liver reserves, most commonly assessed by the Child–Turcotte–Pugh score. HCC is also considered to be a chemotherapy-resistant tumor [27]. No standard therapy existed for advanced HCC until 2008, when the landmark SHARP trial was reported [28] and firmly established sorafenib as the standard of care. The SHARP investigators only included patients with inoperable disease and a Child–Pugh class A status, and compared sorafenib to a placebo. Both overall survival (10.7 vs. 7.9 months) and time to radiologic progression (5.5 vs. 2.8 months) favored the Sorafinib group. Patients with Child–Pugh class B or C liver disease were not included in this trial.

Efforts to improve on first-line sorafenib were initially unsuccessful, and only in the last few years have several other novel agents been approved for HCC. The REFLECT study [29] compared lenvatinib with sorafenib in the first-line setting for unresectable HCC. Almost all (99%) patients had Child–Pugh class A disease. Patients with involvement of >50% of the liver or invasion of the main portal vein or biliary tree were excluded. This was a noninferiority trial and lenvatinib was found to be noninferior to sorafenib for the primary endpoint of median overall survival (13.6 vs. 12.3 months, HR 0.92). The objective response rate was higher (24% vs. 9%) with lenvatinib, and median time to progression (TTP) was longer (7.4 vs. 3.7 months). Until recently, these two drugs represented acceptable first-line options.

Atezolizumab is a monoclonal antibody and immune checkpoint inhibitor that binds to programmed cell death ligand 1 (PD-L1). Atezolizumab in combination with bevacizumab, a vascular endothelial growth factor (VEGF) monoclonal antibody, was compared with sorafinib in the phase III IMbrave 150 trial [30]. In this first-line trial, 501 patients with advanced, unresectable HCC and Child–Pugh class A liver disease were randomly assigned to atezolizumab plus bevacizumab or to sorafenib in a 2:1 ratio. All patients were required to undergo an upper endoscopy within six months of treatment initiation due to the risk of bleeding, and if varices were found, these had to be addressed prior to treatment. In this global, open-label, phase 3 trial, the overall survival at 12 months was 67.2% (95% CI, 61.3 to 73.1) with atezolizumab–bevacizumab and 54.6% (95% CI, 45.2 to 64.0) with sorafenib. The hazard ratio for death with atezolizumab–bevacizumab as compared with sorafenib was 0.58 (95% confidence interval [CI], 0.42 to 0.79; $p < 0.001$). The median progression-free survival of the atezolizumab–bevacizumab group was 6.8 months (95% CI, 5.7 to 8.3) compared to 4.3 months (95% CI, 4.0 to 5.6) with sorafenib. Grades 3 or 4 adverse events were different but occurred in a similar percentage of patients in each group (57% vs. 55%).

This was the first trial to show a survival advantage of a novel treatment regimen over sorafenib. A combination of atezolizumab and bevacizumab is now considered the new standard of care in the first-line setting for appropriate patients with advanced HCC. The benefit is maintained in an updated analysis [31], with a median follow-up of 15.6 months. The median overall survival with combined atezolizumab–bevacizumab therapy was 19.2 months vs. 13.4 months for sorafinib. The objective response rate was also higher with the atezolizumab–bevacizumab combination compared to sorafenib (30% vs. 11%).

There are several upcoming studies evaluating various combination therapies, including the lenvatinib–pembrolizumab study NCT03713593 [32]. Results from this are expected in the coming months, and may yet change our first-line options.

With the adoption of atezolizumab plus bevacizumab in the first-line setting, the previous first-line options of sorafenib and lenvatinib can now be used in the second-line sphere. Randomized clinical data do not exist in this setting, but the group felt that it would be reasonable to use these drugs after atezolizumab–bevacizumab in appropriate patients with good performance status and acceptable liver reserves.

Regorafenib and cabozantinib have been studied in patients who have progressed on first-line treatment with sorafenib. The RESORCE trial [33] randomized 573 patients who had progressed on sorafenib and still had a good performance status (0 and 1) and liver

function (Child–Pugh class A) to regorafenib or placebo. Median overall survival (10.6 vs. 7.8 months, HR for death 0.63), as well as higher rates of objective antitumor response (11% vs. 4%) was seen in favor of regorafenib. Dose modification was required for 68% of the patients receiving regorafenib, but with this dose modification, it was reasonably well tolerated.

The phase III second- or third-line CELESTIAL trial [34] randomized 707 patients with advanced and progressing HCC and Child–Pugh class A cirrhosis to cabozantinib or placebo. The median overall survival was significantly better with cabozantinib (10.2 vs. 8.0 months), and the benefit was more pronounced in patients receiving cabozantinib in the second-line setting (median overall survival 11.3 vs. 7.2 months). It should be noted that about one-third of the patients received cabozantinib in the third-line setting. Most patients had stable disease, and an objective response rate of only 4% was seen with cabozantinib. This led to the approval of cabozantinib in the second- or third-line setting.

For patients who do not receive immunotherapy in the first-line setting, there are scant data to address the use of checkpoint inhibitors in the second-line settings and subsequently. The phase II Keynote-224 trial [35] of pembrolizumab after prior treatment with sorafenib showed an objective response rate of 17% and stable disease in 44% of patients. The median duration of response was 4.2 months. This is further supported by the Keynote 240 trial [36].

In the phase I/II CheckMate 040 study, patients who progressed on sorafinib received nivolumab monotherapy [37]. In this study, 49 of the 255 (19%) patients assessable had an objective antitumor response to nivolumab and the median duration of response was more than 9 months.

Based on these limited data, immunotherapy with a checkpoint inhibitor can be considered in patients who are refractory or intolerant to tyrosine kinase inhibitors and have not received immunotherapy in the first-line setting.

**Author Contributions:** All authors listed contributed to the online collaboration in writing this document. A.Z. was primarily responsible for the review, editing and submission of the document. All authors have read and agreed to the published version of the manuscript.

**Funding:** The 2021 WCGCCC virtual meeting received unrestricted educational grants from Pfizer, Hoffmann-La Roche, AstraZeneca Canada, Roche, Eisai, Bristol Myers Squibb, Merck Inc. and Taiho Canada. During the entire process, the sponsors had no influence whatsoever over the development of the guidelines, and they did not review or read the guidelines before submission. No author was compensated for their work on this article.

**Institutional Review Board Statement:** Not applicable.

**Informed Consent Statement:** Not applicable.

**Data Availability Statement:** Not applicable.

**Acknowledgments:** The WCGCCC organizing committee thanks all meeting participants for their contributions to the development of this consensus statement. In addition, the committee thanks the meeting sponsors and Buksa Strategic Conference Services for support in organizing the meeting.

**Conflicts of Interest:** J.D. has conducting clinical trials for BMS, Merck, MedImmune and Astellas Array BioPharma, and is a consultant for AstraZeneca, Eisai, Taiho and Amgen. C.K. received an unrelated research grant from Celgene Inc. and an honorarium from Amgen. H.L. received honoraria from Merck, BMS, AstraZeneca, Eisai, Taiho, Roche, Amgen and Bayer for consultant work. D.L. is a consultant for Sirtex Medical and AstraZeneca, both a speaker and consultant for Eisai, and a speaker for Ethicon Endosurgery. R.L.-Y. has had advisory roles for Eisai, Ipsen, AstraZeneca, Roche and Celgene. K.M. has an advisory role for Pfizer Canada, Eisai Inc. and Bayer Canada and has received clinical trials funding from Deciphera Pharmaceuticals, BluePrint Medicines and AstraZeneca. D.S. has received grant funding from Varian Medical, honoraria from AstraZeneca, Merck and Pfizer, and is on the advisory board for AstraZeneca. A.Z. has received travel grants from Roche and has also performed consultant work for Eisai. The remaining authors declare no conflicts of interest.

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
