# Peer review of "Report from the Western Canadian Gastrointestinal Cancer Consensus Conference Virtual Education Series—Transition from Local to System Therapy and Optimal Sequencing of Systemic Therapy for HCC"

_curroncol, doi:10.3390/curroncol28060367_

Round 1
Reviewer 1 Report
I agree with all the statements of this consensus conference
the revised version improved the quality of presentation
Reviewer 2 Report
The authors adequately responded to reviewer’s comments. I have no additional comments for further revisions.
This manuscript is a resubmission of an earlier submission. The following is a list of the peer review reports and author responses from that submission.
Round 1
Reviewer 1 Report
This consensus conference focused on the hot topico of novel LRT and systemic therapies for HCC
I suggest to provide anputative flowchart for better decision making
The potential role of LT after effective downstaging with systemic treatments shouldnalso be discussed (10.1111/tri.13983)
Beat regards
Author Response
Flow chart has been added for both local and systemic therapy.
This meeting was the consensus conference and the role of local therapy after downstaging with systemic therapy was not specifically discussed and it would be difficult to add or attribute a specific statement to the meeting at this time. However, I have modified the text to indicate that this could be considered in a multidisciplinary setting

Reviewer 2 Report
This is a paper reporting the discussion in the consensus conference on the treatment of HCC. The two topics regarding when to consider the switch from logo regional therapy to systemic therapy and which regimens to be used as first, second, or third line treatment.
I have several concerns as the follows.
- Although the indication of locoregional treatment would not be the topic of this paper, it was not natural that only the indication of TACE, SBRT, and radioembolization but not that of RFA or hepatectomy were described. Even for extrahepatic disease, surgical resection could be indicated in selected cases. Because the participants of the consensus conference included surgical oncologists, the indication of surgical resection should also be discussed.
- I think the authors' description that patients following "two" successive TACE without objective response should be referred for systemic treatment is just the recommendation by the experts rather than the conclusion based on any clinical studies. Recommendation and evidence conducted from clinical studies should be clearly differentiated.
- A figure such as flowchart will help facilitate the understanding of the recommended choice of each regimen for first, second, and third or fourth line treatment as systemic therapy.
Author Response
Thank you for further review and a thoughtful comments.
- The indications for surgical resection were not specifically discussed during this meeting. This meeting was the consensus conference and it is difficult to attribute or add something to the meeting at this point. However, the text test been modified and additional references added to address the role of surgery.
- The comment about 2 successive TACE is correct. The text has been modified to reflect and highlight this
- Flow chart has been added for both local and systemic therapy
